# Unsung climate guardians: The overlooked role of remnant and spontaneous trees in carbon stocks and gains from tree growth in West African cocoa fields

Isaac Kouamé Konan[1], Anny Estelle N'Guessan[1], Aimé Kouassi[2], Marie Ruth Dago[2], Justin Kassi N'Dja[1], Raphaël Aussenac[2,3,4], Stéphane Guei[5], Patrick Jagoret[6], Bruno Hérault[3,4]*

**1** UPR de Botanique, UFR Bioscience, Université Félix Houphouët-Boigny, Abidjan, Côte d'Ivoire, **2** UMRI SAPT (Sciences Agronomiques et Procédés de Transformation), Institut National Polytechnique Félix Houphouët-Boigny, Yamoussoukro, Côte d'Ivoire, **3** UPR Forêts et Sociétés, CIRAD, Montpellier, France, **4** Forêts et Sociétés, Université Montpellier, CIRAD, Montpellier, France, **5** Centre d'excellence Africain sur le changement climatique la biodiversité et l'agriculture durable, Université Félix Houphouët-Boigny, Abidjan, Côte d'Ivoire, **6** CIRAD, UMR ABSys, Montpellier, France

* bruno.herault@cirad.fr

## Abstract

Cocoa cultivation in West Africa has been a major driver of deforestation, leading to increased greenhouse gas emissions and threatening cocoa yields. Agroforestry, which integrates trees from various origins—remnant, spontaneous, and planted—presents a sustainable solution to enhance carbon sequestration and improve farm resilience. However, the specific contributions of these tree origins and the socio-environmental factors shaping their effectiveness remain poorly understood. This study examines carbon dynamics across 150 cocoa fields in Côte d'Ivoire, analyzing a total of 11,568 trees across 15 sites. Using Bayesian modeling, we assess carbon stocks and gains from tree growth to explore how socio-environmental factors influence carbon balance in cocoa fields. Carbon stocks varied widely with remnant having the highest median carbon stocks (6.33 Mg/ha), followed by spontaneous (2.06 Mg/ha) and planted trees (1.53 Mg/ha). Carbon gains are similar for planted and spontaneous trees up to 7 years, but afterward, spontaneous trees grow faster (11.20 ± 0.87 kg/year) than planted ones (3.96 ± 0.5 kg/year). Carbon stocks rose with informed farmers and former forest use, but declined with higher cocoa density. Carbon gains at the tree level is primarily influenced by ownership and previous forest land use with positive effects, while cocoa density and annual temperature have negative effects. To maximize carbon sequestration and ensure the sustainable management of agroforestry systems, interventions should prioritize securing land tenure, enhancing farmer training in tree botany, and promoting the conservation of remnant and spontaneous trees.

**Data availability statement:** All original data files are available from the DRYAD database repository (https://doi.org/10.5061/dryad.47d7wm3q5).

**Funding:** This study was conducted within the framework of the Cocoa4Future (C4F) project, which is funded by the European DeSIRA Initiative under grant agreement No. FOOD/2019/412-132 and by the French Development Agency. All authors benefited from this funding. The funders had no role in study design, data collection and analysis, decision to publish, or preparation of the manuscript.

**Competing interests:** The authors have declared that no competing interests exist.

## Introduction

In West Africa, the expansion of agricultural land for cocoa cultivation is a major driver of global greenhouse gas (GHG) emissions, largely due to the widespread deforestation it causes [1–3]. In fact, Vervuurt [4] estimate that approximately 1.47 kg of $CO_2$ is released for every kilogram of cocoa produced. This deforestation exacerbates climate change, leading to disruptions in global weather patterns, which, in turn, negatively impact agricultural production, including cocoa. As global temperatures rise and precipitation patterns become more unpredictable, the effects on cocoa yields are becoming increasingly severe [5,6]. This problem is especially critical in areas where cocoa is grown as a monoculture. Without the natural shade provided by diverse vegetation, these monoculture plantations are more vulnerable to the impacts of climate variability [7]. Thus, the interplay between deforestation, GHG emissions, and climate change is creating a feedback loop that threatens not only the environment but also the future of cocoa production in these regions. To address these challenges, agroforestry systems (AFS) have been proposed as a potential solution for both mitigating and adapting to climate change. Several studies [8,9] have demonstrated that integrating a diversity of species into cocoa plantations through agroforestry can provide ecosystem services that enhance the resilience of cocoa farms. These services include creating a microclimate conducive to cocoa growth [10], improving soil fertility through the provision of nutrients, and reducing soil erosion [11,12]. In terms of carbon sequestration, agroforestry systems outperform monocultures, with higher carbon storage capacities [10,13,14]. Dixon [15] highlighted that when properly managed, AFS can thus act as long-term carbon sinks.

Several international initiatives, such as REDD+ (Reducing Emissions from Deforestation and Forest Degradation), the Cocoa Forest Initiative, and the ARS100 standard in West Africa which was recently proposed and implemented by Côte d'Ivoire and Ghana, aim to promote the adoption of AFS in cocoa production systems [16,17]. Organizations like Rainforest Alliance, UTZ Certified, and Fairtrade also advocate for agroforestry practices in the cocoa sector [18,19]. Despite ongoing efforts, progress has been hindered by an overemphasis on only tree planting, often at the expense of promoting natural regeneration [20]. Recent studies [20,21] have indeed highlighted the coexistence of three distinct types of tree cohorts within cocoa AFS: (1) remnant trees, preserved during land clearing; (2) spontaneous trees, which regenerate naturally during or after cocoa field establishment; and (3) planted trees, intentionally introduced by farmers. Preliminary evidence suggests that remnant and spontaneous trees could have higher species diversity and potentially greater carbon sequestration capacity compared to planted trees [20,22]. However, despite the increasing interest in carbon stocks and gains in cocoa plantations, research remains limited due to the influence of various socio-environmental factors [9,23]:

- Climatic conditions, particularly precipitation and temperature, are crucial for carbon dynamics in cocoa plantations. Adequate rainfall supports tree growth and photosynthesis, while insufficient precipitation can slow growth and reduce carbon uptake [24–26]. Excessive rainfall may lead to soil erosion and nutrient loss.

Additionally, higher temperatures can slow carbon accumulation due to trees' physiological responses that limit evapo-transpiration [27,28].

- Social factors, including farmers' knowledge of the environment and land ownership, influence carbon storage. Farmers with a strong understanding of local ecosystems are more likely to implement effective management practices that enhance carbon dynamics [21,29,30]. Non-owner farmers might prioritize short-term gains, impacting their long-term sustainability practices.

- Field characteristics, such as cocoa planting density and previous land use, also affect carbon dynamics. High planting density increases competition with associated trees, limiting biomass and carbon sequestration [10,31]. In contrast, lower density allows for greater biodiversity and carbon storage. The previous land use is also significant; fields converted from forests typically retain higher carbon stocks compared to those converted from other agricultural uses [9,32].

- Soil characteristics, particularly organic matter and bulk density are vital for aboveground carbon storage potential. High organic matter content enhances soil health and carbon sequestration in trees [33]. Conversely, high bulk density indicates soil compaction, which restricts root growth and limits the soil's ability to retain water and nutrients. Healthy soils not only support robust tree growth but also enhance the ability of these systems to sequester carbon [34].

To date, no comprehensive study has examined the combined impact of socio-environmental factors on carbon stocks and gains in trees within cocoa agroforestry systems. To fill this gap, the present study aims to identify the factors influencing the carbon sequestration capacity of trees in cocoa farms, with the goal of enhancing their role as sustainable terrestrial carbon sinks. Specifically, this research seeks to address the following questions: (1) At the cocoa field level, what are the carbon stocks of planted, spontaneous, and remnant trees? (2) At the individual tree level, what is the effect of origins (planted, spontaneous) on the carbon gains from tree growth? (3) What are the key socio-environmental factors influencing carbon stocks (at the field level) and gains (at the tree level) in these systems?

## Materials and methods

### Study sites

Study plots were selected from 15 sites across the cocoa-producing forest zone of Côte d'Ivoire (Fig 1). The selection aimed to maximize three key gradients: (i) the North-South climatic gradient, with mean annual temperatures ranging from 22.6°C to 26.2°C and annual rainfall varying from 1,900 mm to 1,100 mm, (ii) the historical gradient of cocoa introduction, from the oldest plantations in the East to more recent ones in the West, and (iii) the forest type gradient, transitioning from evergreen forests in the South to semi-deciduous forests further North. Rather than a systematic sampling approach, site selection was opportunistic, based on our field expertise and prior knowledge of specific zones. This approach allowed us to capture a diverse range of environmental and historical contexts while leveraging existing insights from farmers and cooperatives.

To capture the diversity of cocoa fields, we identified 10 cocoa fields at each site, focusing on three key criteria: (i) structural complexity, ranging from nearly full-sun cocoa monocultures to complex cocoa agroforests; (ii) age of the cocoa fields, encompassing both young (< 20 years) and mature (> 20 years) plots; and (iii) cocoa tree yield, varying from low productivity (< 300 kg/ha) to high productivity (> 300 kg/ha). The study plots, covering a total area of 240.5 ha, vary in size from 0.3 to 5 hectares and serve as the sampling units, corresponding to the management units of the farmers.

### Ethic statement

The objectives of this study were presented to each village chief, and permission was requested and granted to conduct surveys in their respective villages. Before each interview, the study's objectives and ethical guidelines for data handling were explained to cocoa farmers and their families. This ensured they understood that they had (i) the right to withdraw

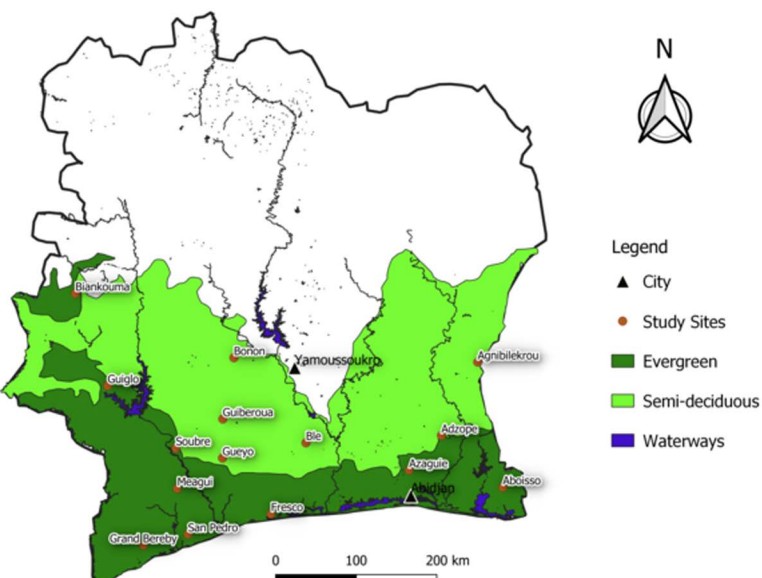

**Fig 1. Map of the 15 study sites illustrating the diversity of forest types and annual precipitation across Ivory Coast, West Africa.** National boundaries and waterways reprinted from OpenStretMap under a ODC-ODbL license.

from the study at any time, (ii) the freedom to refuse to answer any question without providing justification, and (iii) the ability to pause the field work as needed. Farmers who verbally confirmed their consent and understanding then signed a declaration outlining the ethical guidelines and authorizing the use of personal data (e.g., knowledge, ownership) and survey responses. All collected data were later anonymized following a pre-established data anonymization procedure. Tree inventory and measurements were primarily conducted in French by the study's lead author, with assistance from one or two interpreters for the local language. Each fieldwork session with a producer lasted an average of six hours.

## Data collection

All cocoa fields were mapped using GPS technology for precise delineation. Each associated tree with a minimum diameter at breast height (dbh) of 10 cm was identified to the species level. The dbh and the height, measured with a Sunto dendrometer, of each tree was recorded. The origin of each tree—classified as planted, remnant, or spontaneous— and the age of each planted and spontaneous tree was recorded based on the farmer's declaration.

Eight socio-environmental factors of interest were identified for this study (Table 1):

- Two social factors related to the cocoa farmer: botanical knowledge and land ownership status of the cocoa field, obtained through interviews with the farmers. The farmers' botanical knowledge was assessed using an image-based questionnaire on a digital tablet. This questionnaire presented three images of each of the 23 most frequent tree species found in Ivorian cocoa fields [35]. Semi-quantitative scores were assigned based on the level of recognition: 0 if the species was not recognized, 1 if the species was recognized but not named in French or local language, and 2 if the species was recognized and correctly named in French and/or the local language. All scores were then summed to obtain an overall assessment of the farmer's knowledge level.

- Two soil factors: organic carbon content and bulk density of the soil, compiled from existing databases (https://soilgrids.org/).

- Two field-related factors: Cocoa tree density was estimated by counting the number of cocoa trees within a 1,000 m² area in each cocoa field. This area was subdivided into five subplots, which were evenly distributed across the field to

**Table 1. List of the 8 socio-environmental factors studied to explain the variability of carbon stocks and gains from tree growth in the 150 cocoa fields.**

| | Name | Description | Unit | Source | Hypotheses |
|---|---|---|---|---|---|
| **Social** | Farmer ownership | the ownership status of the farmer's cocoa (owner or not owner) | Binary | Interviews with farmers | Ownership may encourage long-term management, increasing tree stocks and gains from tree growth |
| | Farmers's tree knowledge | Recognition test of the 23 most common species in agroforestry | % | Interviews with farmers | Greater knowledge may lead to higher tree diversity, boosting both stocks and gains from tree growth |
| **Field** | Previous land-use | Previous landscape (forest or not) | Binary | Interviews with farmers | Former forests may retain higher tree stocks and support faster regeneration, influencing gains from tree growth |
| | Cocoa density | Cocoa tree density | Ind/ha | Measured on 1000 m² square | Higher cocoa density may limit space for other trees, reducing stocks and slowing gains from tree growth |
| **Soil** | Bulk density | Bulk density from a depth of 30–60 cm | Kg/dm³ | https://soilgrids.org/ | Higher bulk density may restrict root growth, lowering stocks and reducing gains from tree growth |
| | Soil organic carbon content | Soil organic carbon content (fine soil fraction) | g/kg | https://soilgrids.org/ | Richer soils may support higher tree biomass, enhancing both stocks and gains from tree growth |
| **Climate** | Annual temperature | Annual temperature from 1979 to 2013 at 1 km resolution | Deg C | https://chelsa-climate.org/ | Higher temperatures may accelerate biomass turnover, increasing gains from tree growth but not necessarily stocks |
| | Annual precipitation | Annual precipitation from 1979 to 2013 at 1 km resolution | mm | https://chelsa-climate.org/ | More rainfall may promote tree growth, increasing both stocks and gains from tree growth |

ensure a representative sampling of topography, tree density, and management history. Previous land use was obtained through interviews with the farmers.

- Two climate factors: annual temperature and precipitation, compiled from existing databases (https://chelsa-climate.org/).

  All original data used in this study are publicly available (https://doi.org/10.5061/dryad.qfttdz0sm).

## Data analysis

The aboveground biomass of each individual tree was estimated using the computeAGB() function from the BIOMASS package [36] in R, then converted to carbon by applying the conversion factor of 0.47 as recommended by Martin and Thomas [37]. The computeAGB() function uses the pantropical allometric equation of Chave [38] to estimate the aboveground biomass, incorporating wood density values obtained via the getWoodDensity() function—matched at the species level when available, or at the genus level otherwise. We evaluated the effects of cohorts (planted, remnant, spontaneous) and 8 socio-environmental *se* factors on (i) Carbon stock (at the plot scale) and (ii) Carbon gains (at the individual scale, considering that each individual has a different age) using lognormal likelihood models. This likelihood was chosen because the response variables are defined on $R^+$.

Carbon stocks were modelled with the following equation

$$Cs_{coh,p} \sim LN[log(\theta_{coh} \times e^{(\sum_{se=1}^{8}(\theta_X \times X_{se,p}))}), \sigma_{Cs}^2] \tag{1}$$

Where $Cs_{coh,p}$ is the observed Carbon stock (Mg) of cohort *coh* in plot *p*, $\theta_{coh}$ is the predicted average carbon stock (Mg) of cohort *coh* in average environmental conditions (*i.e.* when $X_{se,p}$ equal zero), $\theta_X$ are the standardized (*i.e.* centred and reduced) effects of the socio-environmental factors and $X_{se,p}$ are the observed values of the socio-environmental factors *se* in a given plot *p*. The effects of the socio-environmental factors are included into an exponential function to guarantee positive predictions of $\theta_{coh}$.

Given that the evolution of the tree carbon over time is well approximated by a power law because growth rates scale with the tree's size, following geometric and metabolic constraints [39], tree carbon gains in time were modelled with the following equation

$$Cf_{i,sp,coh,p} \sim LN[log(\theta_{coh} \times \theta_{sp} \times Age_i^{\beta_{coh}} \times e^{(\sum_{se=1}^{8}(\theta_X \times X_{se,p}))}), \sigma_{Cf}^2]$$

(2)

$$\theta_{sp} \sim LN(log(1), \sigma_{sp}^2)$$

(3)

Where $Cf_{i,sp,coh,p}$ is the observed carbon of tree $i$ from cohort $coh$ of species $sp$ in plot $p$. $\theta_{coh}$ captures the cohort-specific baseline effect on carbon gain (kg.year$^{-1}$), while $\theta_{sp}$ represents the species effect, with a lognormal prior centered at 0 to account for interspecific variability. The exponent $\beta_{coh}$ is the power law coefficient that defines how carbon gains scale with age for each cohort, allowing the age-growth relationship to very across planted and spontaneous trees. This flexibility reflects ecological and management-driven differences in growth patterns, in line with the general shape predicted by metabolic theory, which informs model structure rather than fixed parameter values.

Socio-environmental factors $X_{se,p}$ at the plot level are included as standardized (centered and scaled) covariates with corresponding effects $\theta_X$, embedded within an exponential function to ensure positive predictions and multiplicative influence on carbon gains. These covariates capture broader landscape, management, and tenure effects but do not interact directly with age or cohort effects. As the age of remnant trees is inherently unknown, the carbon gain model is applied only to planted and spontaneous trees, for which age estimates are available. Furthermore, because very few trees exceed 40 years of age, the analysis of growth-derived carbon gains is effectively restricted to younger trees, ensuring more reliable age estimates and minimizing uncertainty related to farmer recall.

All models were implemented in R and inferred within a Bayesian framework using Stan, which relies on the Hamiltonian Monte Carlo (HMC) method to achieve full Bayesian inference [40]. Bayesian methods provide full posterior distributions of parameters rather than point estimates, enabling a more comprehensive quantification of uncertainty. This is especially relevant for modelling carbon stocks and gains from tree growth, which are subject to complex environmental and demographic influences. The use of HMC via Stan allows for efficient exploration of high-dimensional parameter spaces, reducing issues such as slow convergence and autocorrelation, which are common in traditional Markov Chain Monte Carlo (MCMC) approaches.

However, it is important to note a key limitation of the modelling framework: while we propagate uncertainty related to model parameters and observed covariates, we are not able to account for potential errors in farmer-reported tree age, which likely introduces additional, unquantified uncertainty into our estimates of carbon gains.

## Results

A total of 11,568 trees were inventoried from 277 species (Appendix 1). Spontaneous trees were the most represented cohort, with 4,734 individuals (41%), followed by planted trees with 3,520 individuals (30%), and finally remnant trees with 3,304 individuals (28%). The remnant tree cohort naturally includes a higher proportion of large trees, with 25% having diameters exceeding 50 cm and 66% reaching heights over 25 m. In contrast, the spontaneous tree cohort comprises 15% of trees with diameters greater than 50 cm and 45% with heights exceeding 25 m. The planted tree cohort contains a very small proportion of large trees, with only 4% having diameters over 50 cm and 32% reaching heights greater than 25 m (Appendix 2 and 3). Specific wood densities did not differ significantly (Kruskall-Wallis test, $P = 0.35$) between cohorts with medians being equal to 0.56, 0.54, 0.55 for, respectively, remnant, spontaneous and planted trees (Appendix 4).

## Carbon stocks

Carbon stocks per hectare range from 0.26 to 137.39 Mg, with a median of 13.81 Mg. The carbon stock values by cohort reveal higher carbon stocks for remnant trees, in contrast to the lower stocks observed in planted and spontaneous trees (Table 2, Fig 2). For remnant trees, carbon stocks per plot range from 0.01 to 86.73 Mg/ha, with a median of 6.33 Mg/ha. Meanwhile, carbon stocks for spontaneous and planted trees range from 0.00 to 137.39 Mg/ha, with a median of 2.06 Mg/ha, and from 0.03 to 13.26 Mg/ha, with a median of 1.53 Mg/ha, respectively.

## Carbon gains

The carbon accumulation curves show significant differences between planted and spontaneous trees (Table 2, Fig 3). Biomass accumulation for both tree types is almost identical between 1 and 7 years, with an average accumulation of 6.0±0.89 kgC and 8.4±0.9 kgC per year respectively for planted and spontaneous trees. However, the accumulation diverges after 7 years, with planted trees accumulating biomass more slowly (3.8±0,6 kgC per year) while spontaneous tree has 11.4±1.7 kgC per year. At 40 years, an average spontaneous tree will have a carbon stock of 471.2 kgC, while a planted tree will only have 200.98 kgC. Therefore, the difference in carbon gain is substantial and increases with age for spontaneous and decreases for plantes trees.

**Table 2. Per cohort descriptive statistics (quantiles 5%, 50%, 95%) of carbon stocks and gains from tree growth.**

| | Stocks (Mg/ha) | | | Gains (kg/tree) < 7 years | | | Gains (kg/tree) >7 years | | |
|---|---|---|---|---|---|---|---|---|---|
| | 5% | 50% | 95% | 5% | 50% | 95% | 5% | 50% | 95% |
| Remnants | 0.40 | 6.33 | 15.99 | | | | | | |
| Spontaneaous | 0.04 | 2.06 | 6.24 | 7.28 | 8.39 | 8.82 | 9.66 | 11.37 | 11.85 |
| Planted | 0.08 | 1.53 | 3.42 | 5.36 | 6.02 | 6.63 | 3.41 | 3.81 | 4.27 |

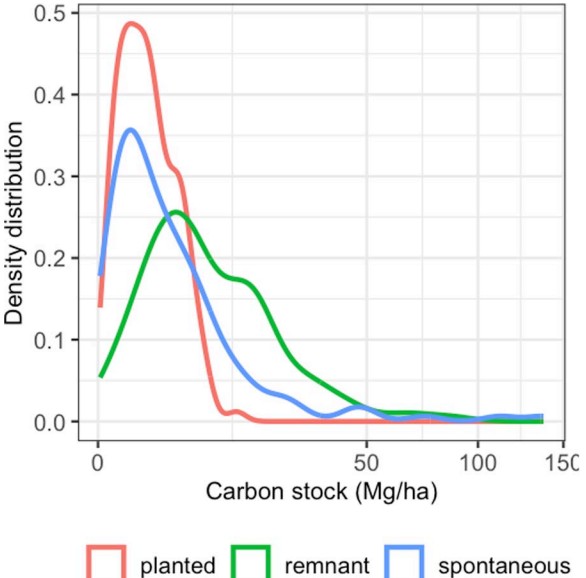

**Fig 2. The distribution of carbon stocks across 150 cocoa fields is shown, with carbon stock on the x-axis and the density of values on the y-axis.**

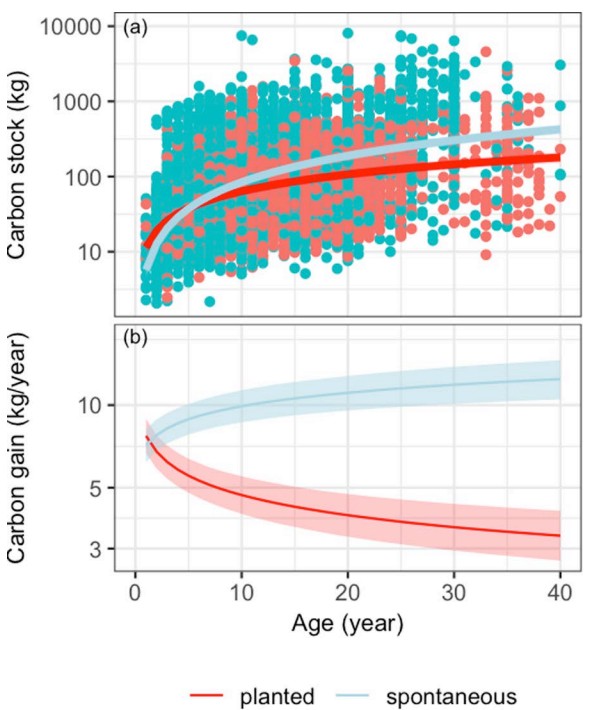

**Fig 3. Carbon accumulation of planted and spontaneous trees.** The points represent the observed carbon stock of trees, while the lines show the predicted temporal dynamics of carbon stock (a) and gain (b), with the 95% credibility intervals shaded. This figure is represented on a logarithmic scale.

## Effects of socio-environmental factors

Carbon stock values of the studied plots are mainly influenced by, in decreasing order, the previous land-use, the ownership status of the cocoa farmer, and the cocoa tree density (Fig 4). These are the factors with the most absolute parameter values different from zero after model inference. Being the owner of the plot ($\theta = -0.39 \pm 0.1$) and having plots with high cocoa tree density ($\theta = -0.34 \pm 0.09$) have negative effects on carbon stock, while the forest previous land-use ($\theta = 0.41 \pm 0.18$) has a positive effect on the tree carbon stock of cocoa plantations.

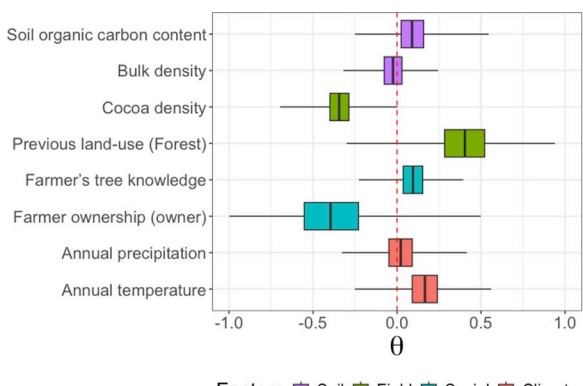

**Fig 4. Effect of social, field, soil, and climate factors on plot-level carbon stock.** All covariates were standardized, allowing for direct comparison of their relative effects.

Carbon gain at the tree level within cocoa plantations is primarily influenced by, in decreasing order, the cocoa farmer's ownership status, the previous land-use, cocoa density and annual temperature (Fig 5). Both the ownership of the plot by the cocoa farmer and the previous land-use (when it's forest) have a positive effect on carbon sequestration by individual trees ($\theta = 0.19 \pm 0.04$ and $\theta = 0.13 \pm 0.03$, respectively). In contrast, both annual temperature and cocoa density negatively affect carbon sequestration at the tree level, with estimated parameters of $\theta = -0.08 \pm 0.01$ and $\theta = -0.13 \pm 0.01$, respectively.

## Discussion

This study helps better understand the carbon sequestration mechanisms in cocoa-based agroforestry systems, and thus provides a foundation for developing management strategies to combat climate change. The objective of this study was to identify the sources of carbon stock and gain in trees within cocoa fields in West Africa, with a focus on Côte d'Ivoire. The carbon stock in these fields is primarily derived from remnant trees, while the stocks from planted and spontaneous trees are much lower. In contrast, for carbon gains, more carbon is accumulated in spontaneous trees than in planted trees. Several socio-environmental factors explain the levels of carbon stock and gains for each tree cohort.

We acknowledge the potential limitation in determining spontaneous tree age and its impact on carbon gain estimates. However, tree selection in cocoa farms occurs in waves, creating age homogeneity that minimizes underestimation bias. Additionally, farmers often link tree age to personal or societal events, providing reliable memory anchors for accurate estimation. While some uncertainty remains, these factors reduce the risk of overestimated carbon gains. Future studies could complement farmer estimates with growth ring analysis or dendrometer measurements to refine age assessments.

We also recognize a limitation regarding the interpretation of farmer botanical knowledge. While our findings show a positive association between farmer knowledge and higher tree carbon stocks and gains, causality may be questioned. Landscapes with higher tree diversity—such as former forest areas—may naturally promote greater knowledge through prolonged exposure, and migration patterns may further complicate the relationship.

### Carbon giants: the remnants

Our results show that carbon stock in cocoa agroforestry systems (CAFs) is strongly influenced by the presence of remnant trees (Fig 2). This finding is consistent with previous studies (e.g. Andreotti [41], Bastin [42], and Sanial [21]), which have also reported that remnant trees contribute disproportionately to aboveground carbon storage compared to planted or spontaneously regenerated cohorts. Remnant trees are typically large, mature individuals that were spared during

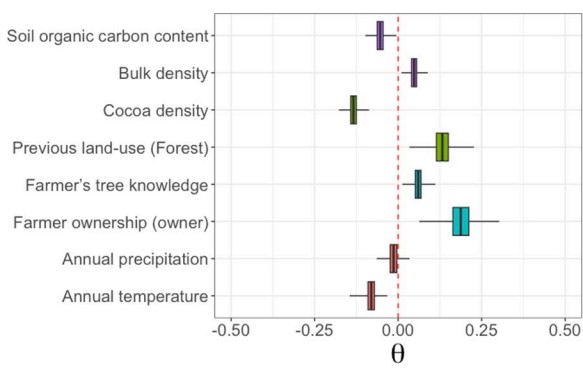

**Fig 5. Effect of soil, field, social, and climate factors on tree-level carbon gain.** All covariates were standardized, enabling direct comparison of their relative effects.

land clearing—either intentionally or due to logistical constraints [43]. They often include species such as *Milicia excelsa* (Welw.) Benth, *Alstonia boonei* De Wild, *Amphimas pterocarpoides* Harms and *Petersianthus macrocarpus* (P. Beauv.) Liben (Table 3), which have high carbon storage because of their large diameter and height [21]. As expected, tree size directly determines carbon stock, with taller and thicker trees storing significantly more carbon [22]. While it could been hypothesized that farmers may selectively leave certain species—such as those with high wood density—because they are difficult to cut, our results show no significant differences in wood density among cohorts (Appendix 4). In addition to size, species diversity may also contribute to higher carbon stocks. The remnant cohort exhibit the highest species among the three groups [21,43], and previous research has shown that greater tree diversity is often associated with increased biomass and carbon storage [22,32,42]. Thus, the value of remnant trees lies not only in their individual size but also in the structural and ecological diversity they bring to these landscapes. In contrast, the planted tree cohort, mainly consisting of small fruit tree species with low diameters, stores the least carbon. These trees include species such as *Citrus sinensis* (L.) Osbeck, *Psidium guajava* Linn, and *Anacardium occidentale* Linn (Table 3).

The ownership status of the cocoa farmer is one of the most significant factors influencing the carbon stock in cocoa fields. A lower carbon stock is observed when the field is managed by the owner, whereas a higher carbon stock is found in fields managed by non-owners. Cocoa farmers who own the land have full rights to manage it, allowing them to prioritize short-term economic gains from cocoa cultivation [44,45]. With the freedom to make decisions, they are more likely to remove large associated trees to maximize space and resources for cocoa trees (indeed, cocoa density is also negatively linked to tree carbon stock, Fig 4), which they view as the primary source of income [29]. This ownership gives them the incentive to clear trees that may be seen as competing for nutrients, water, and light. In contrast, non-owners, such as sharecroppers or tenants, have limited rights to modify the land and typically may not have the authority to remove trees, especially if doing so requires the landowner's approval. Non-owners may also value more the ecological benefits of large trees [29], which could discourage them from clearing the land.

Our results suggest that carbon stocks are higher in areas with a history of forest land use and that the level of botanical knowledge among cocoa farmers is positively associated with higher tree carbon stocks (Fig 4). Former forest landscapes tend to support a greater number of remnant trees, which are key in shaping farmer knowledge. These trees, often preserved from past forests, provide long-term ecological, economic, and cultural benefits, and farmers who interact with them over time develop a deeper understanding of their value.

Farmers with greater botanical knowledge seem less inclined to cut trees down, likely because they recognize their diverse benefits beyond timber or immediate financial gain [21]. For example, remnant trees serve as natural barriers against soil erosion, improve soil fertility through nitrogen fixation, and provide essential shade for cocoa plants, helping to maintain optimal growing conditions [12,20]. As these trees persist in the landscape, they become living sources of knowledge, teaching farmers about their multifunctional roles. This knowledge fosters more sustainable practices which enhance carbon sequestration. Additionally, farmers who understand the value of trees for craft, fruit, or medicinal uses may be more likely to preserve them [29]. This is particularly important as these resources are disappearing in many landscapes due to deforestation [46,47]. As forests are cleared for agricultural expansion, valuable tree species that provide materials for crafts, fruits for consumption, or medicinal properties are lost, along with their cultural and economic benefits. However, where remnant trees persist, they not only contribute to carbon storage but also help maintain a knowledge base that encourages conservation.

Ultimately, these results suggest that previous forest land use supports the persistence of remnant trees, which in turn shape farmer knowledge. This increased botanical awareness promotes practices that optimize both carbon storage and sustainable land management.

## Carbon hoarders: The spontaneous

The results obtained show a significant difference in carbon sequestration capacity between planted and spontaneous trees within cocoa agroforestry systems. Spontaneous trees, which emerge through natural regeneration, exhibit a higher annual

**Table 3. Top 30 Species by Carbon Stocks for Each Tree Origin (Remnant, Spontaneous, and Planted) in the 150 Studied Fields (240.5 ha).**

| Remnants | | | | Spontaneous | | | | Planted | | | |
|---|---|---|---|---|---|---|---|---|---|---|---|
| Species | Family | N | Total carbon stock (Mg) | Species | Family | N | Total carbon stock (Mg) | Species | Family | N | Total carbon stock (Mg) |
| *Ceiba pentandra* | Malvaceae | 99 | 402.9 | *Elaeis guineensis* | Arecaceae | 993 | 233.8 | *Persea americana* | Lauraceae | 852 | 125.7 |
| *Milicia excelsa* | Moraceae | 85 | 211.4 | *Ceiba pentandra* | Malvaceae | 47 | 84.8 | *Mangifera indica* | Anacardiaceae | 377 | 121.6 |
| *Alstonia boonei* | Apocyna-ceae | 110 | 129.8 | *Alstonia boonei* | Apocyna-ceae | 58 | 80.1 | *Cola nitida* | Malvaceae | 321 | 58.4 |
| *Bombax buonopozense* | Malvaceae | 44 | 107 | *Pycnanthus angolensis* | Myristica-ceae | 84 | 53.6 | *Citrus sinensis* | Rutaceae | 602 | 33.3 |
| *Pycnanthus angolensis* | Myristica-ceae | 81 | 96.9 | *Ricinodendron heudelotii* | Euphor-biaceae | 49 | 44.6 | *Hevea brasiliensis* | Euphorbiaceae | 290 | 31.6 |
| *Elaeis guineensis* | Arecaceae | 229 | 93.2 | *Morinda lucida* | Rubiaceae | 144 | 43.9 | *Terminalia superba* | Combretaceae | 120 | 21.2 |
| *Antiaris toxicaria* | Moraceae | 69 | 77.1 | *Albizia zygia* | Fabaceae | 98 | 40.2 | *Elaeis guineensis* | Arecaceae | 51 | 16.1 |
| *Cola gigantea* | Malvaceae | 29 | 66.4 | *Amphimas pterocarpoides* | Fabaceae | 63 | 39.6 | *Cocos nucifera* | Arecaceae | 138 | 15.1 |
| *Ricinodendron heudelotii* | Euphor-biaceae | 62 | 55 | *Spathodea campanulata* | Bignonia-ceae | 39 | 39 | *Tectona grandis* | Lamiaceae | 95 | 12.8 |
| *Amphimas pterocarpoides* | Fabaceae | 64 | 54.8 | *Milicia excelsa* | Moraceae | 69 | 37.1 | *Petersianthus macrocarpus* | Lecythidaceae | 1 | 6 |
| *Entandrophragma angolense* | Meliaceae | 57 | 40 | *Antiaris toxicaria* | Moraceae | 95 | 34.9 | *Terminalia ivorensis* | Combretaceae | 14 | 4.8 |
| *Albizia adianthifolia* | Fabaceae | 52 | 38.8 | *Sterculia tragacantha* | Malvaceae | 71 | 32.7 | *Garcinia kola* | Clusiaceae | 6 | 4.3 |
| *Mangifera indica* | Anacardi-aceae | 76 | 37.8 | *Blighia sapida* | Sapinda-ceae | 7 | 32.3 | *Anacardium occidentale* | Anacardiaceae | 188 | 4.1 |
| *Terminalia superba* | Combreta-ceae | 47 | 36.7 | *Cecropia peltata* | Urticaceae | 124 | 30.7 | *Spondias mombin* | Anacardiaceae | 32 | 4 |
| *Morinda lucida* | Rubiaceae | 96 | 36.6 | *Ficus exasperata* | Moraceae | 297 | 26.6 | *Gmelina arborea* | Lamiaceae | 14 | 3.3 |
| *Petersianthus macrocarpus* | Lecythida-ceae | 15 | 35.2 | *Terminalia superba* | Combreta-ceae | 82 | 25.7 | *Citrus muricata* | Rutaceae | 44 | 3.2 |
| *Hannoa klaineana* | Simarou-baceae | 31 | 33.9 | *Discoglypremna caloneura* | Euphor-biaceae | 40 | 22.5 | *Milicia excelsa* | Moraceae | 9 | 3.2 |
| *Cola nitida* | Malvaceae | 136 | 33.7 | *Zanthoxylum gilletii* | Rutaceae | 13 | 22.1 | *Citrus maxima* | Rutaceae | 42 | 3.1 |
| *Piptadeniastrum africanum* | Fabaceae | 23 | 31.2 | *Lannea welwitschii* | Anacardia-ceae | 44 | 21.9 | *Cedrela odorata* | Meliaceae | 10 | 2.8 |
| *Albizia zygia* | Fabaceae | 69 | 29.2 | *Mangifera indica* | Anacardia-ceae | 88 | 21.2 | *Psidium guajava* | Myrtaceae | 55 | 2.5 |
| *Sterculia tragacantha* | Malvaceae | 64 | 28.1 | *Petersianthus macrocarpus* | Lecythida-ceae | 23 | 20.7 | *Artocarpus communis* | Moraceae | 15 | 2.5 |
| *Zanthoxylum gilletii* | Rutaceae | 29 | 27.1 | *Bombax buonopozense* | Malvaceae | 19 | 18 | *Tieghemella heckelii* | Sapotaceae | 1 | 2.3 |
| *Piptadeniastrum africana* | Fabaceae | 7 | 22 | *Cola nitida* | Malvaceae | 84 | 17.1 | *Ricinodendron heudelotii* | Euphorbiaceae | 19 | 2.3 |
| *Bombax brevicuspe* | Malvaceae | 8 | 21.6 | *Strombosia pustulata* | Olacaceae | 18 | 16.9 | *Irvingia gabonensis* | Irvingiaceae | 10 | 2.2 |
| *Celtis zenkeri* | Cannaba-ceae | 26 | 21.5 | *Spondias mombin* | Anacardia-ceae | 133 | 16.7 | *Blighia sapida* | Sapindaceae | 6 | 2.1 |

*(Continued)*

**Table 3.** (Continued)

| Remnants | | | | Spontaneous | | | | Planted | | | |
|---|---|---|---|---|---|---|---|---|---|---|---|
| Species | Family | N | Total carbon stock (Mg) | Species | Family | N | Total carbon stock (Mg) | Species | Family | N | Total carbon stock (Mg) |
| *Margaritaria discoidea* | Phyllanthaceae | 48 | 21.5 | *Dracaena mannii* | Asparagaceae | 21 | 16.2 | *Citrus reticulata* | Rutaceae | 58 | 1.9 |
| Ficus exasperata | Moraceae | 136 | 18.5 | Bombax brevicuspe | Malvaceae | 13 | 15.7 | Antiaris toxicaria | Moraceae | 1 | 1.1 |
| Ficus mucuso | Moraceae | 31 | 18.3 | Entandrophragma angolense | Meliaceae | 52 | 15.5 | Acacia mangium | Fabaceae | 15 | 1 |
| Spathodea campanulata | Bignoniaceae | 29 | 17.8 | Margaritaria discoidea | Phyllanthaceae | 43 | 14.9 | Morinda lucida | Rubiaceae | 5 | 1 |

biomass accumulation rate compared to planted trees (Fig 3). It is uncommon for these two origins to be compared in terms of carbon gains at tree level. However, studies conducted on two Pinus species (*Pinus contorta* and *Pinus sylvestris*) in temperate forest ecosystems demonstrate that planted trees can accumulate more biomass than naturally regenerated trees, provided they grow under optimal conditions with selected clones [48,49]. In CAFs, such optimal conditions are not met, as trees are often wild and non-selected, whether planted or spontaneous [20]. Two main factors could explain the difference in biomass accumulation rates between planted and spontaneous trees in cocoa systems. The first factor relates to adaptation to biophysical and climatic conditions. Spontaneous trees, emerging through the process of natural regeneration [34,50], benefit from high genetic diversity, which provides them with a greater predisposition to local conditions [51]. This diversity, shaped by natural selection, allows spontaneous trees to adapt more efficiently to biophysical constraints, such as soil type and moisture, as well as climatic stresses like droughts and heatwaves [52,53]. The second factor concerns the quality and morphology of root systems. Planted trees often suffer from root damage during their transfer from nurseries to plantations or when they are uprooted and moved from one location to another [48]. Such damage primarily affects two types of root systems. Taproots, which are critical for anchorage and access to deep water, are frequently cut or deformed during transplantation, compromising the tree's ability to extract the resources necessary for growth [54]. Trees with damaged taproots are more vulnerable to drought and display slower growth. Superficial roots, on the other hand, may develop abnormally in nurseries due to inappropriate containers or root sleeves, leading to spiralized, compressed, or malformed roots [55]. These deformities inhibit nutrient and water absorption, thereby limiting growth and carbon sequestration capacity. In contrast, spontaneous trees, which recruit naturally on-site, develop intact root systems that are well adapted to local conditions. This promotes better root activity and, consequently, faster biomass accumulation [56].

This study highlights the socio-environmental factors influencing carbon gains in planted and naturally regenerated trees within cocoa agroforestry systems, identifying three main drivers: the land tenure status of the cocoa farmer, annual temperature, cocoa tree density and previous land-use (Fig 5). The carbon gain at the tree level is positively influenced by farmers who own their plots, whereas it is negatively affected by non-owner farmers. As previously shown, owner farmers tend to remove remnant trees from their fields (Fig 4), which negatively impact carbon stocks. At first glance, the tendency of owner farmers to eliminate remnant trees while encouraging the growth of naturally regenerated ones may seem contradictory. However, the explanation lies in the differing roles and perceptions of these trees [29]. Owner farmers actively select and retain naturally regenerated trees for the services they provide, while remnant trees are often tolerated or endured rather than deliberately chosen [57]. This selective approach means that naturally regenerated trees benefit from favorable treatment, as owner farmers tend to nurture them to maximize their expected benefits, such as shade for cocoa, soil improvement, or additional income from fruits and timber. In contrast, tenant farmers, who may lack long-term land security, are less likely to invest in tree management, leading to fewer naturally regenerated trees being retained

and lower carbon gain from tree growth. Furthermore, land ownership grants farmers greater autonomy in making long-term decisions about tree conservation and agroforestry practices. Owners are more inclined to adopt sustainable land management strategies, such as protecting young seedlings, pruning to enhance growth, and integrating naturally regenerated trees into their farming systems. As a result, these trees experience normal or even accelerated growth due to the continuous care provided by owner farmers. This behaviour explains the positive effect of land ownership status on the carbon gain of both planted and naturally regenerated trees. By contrast, in plots managed by tenant farmers, restrictions imposed by landlords or short-term tenure arrangements may discourage investment in tree maintenance, leading to lower regeneration rates and reduced carbon gains. Ultimately, secure land tenure fosters an environment where natural regeneration is actively encouraged, contributing to higher biomass accumulation and enhanced carbon sequestration.

Three additional key results shed light on the factors modulating carbon gains at the tree level in cocoa agroforestry systems. (i) Carbon gain from tree growth diminishes with increasing cocoa tree density, likely due to competition for resources such as light, water, and nutrients. High tree density can lead to overlapping root systems and shaded canopies, which limit the growth potential of individual trees and reduce overall carbon uptake [48]. This finding underscores the importance of optimizing tree spacing in agroforestry systems to balance productivity and carbon sequestration. (ii) Carbon gain from tree growth increases with previous forest land use, likely due to higher soil fertility in areas with a history of forest cover. Forest soils often retain greater organic matter and nutrient content compared to soils converted from other land uses such as grasslands or croplands. This fertility advantage promotes vigorous growth in planted and naturally regenerated trees, enhancing their capacity to sequester carbon. These results emphasize the role of land-use history in shaping the biophysical conditions [41] that support carbon gain from tree growth in cocoa agroforestry systems. (iii) As for annual temperature, negative effects on tree biomass accumulation were observed. Our results are somewhat aligned with several studies that show a negative effect of rising annual temperatures on tree biomass accumulation [27]. The temperature effect is consistent and clear: in response to rising temperatures, trees reduce their photosynthetic activity by closing their stomata to limit evapotranspiration [58,59]. This results in a decrease in biomass production and, consequently, biomass accumulation.

## Conclusions

This study provides a detailed assessment of carbon dynamics in cocoa agroforestry systems by examining carbon stocks and tree-level carbon gains, while also identifying the key socio-environmental factors that influence these outcomes.

(1)  At the field level, we found that remnant trees store the largest share of aboveground carbon, far surpassing that of planted and spontaneous trees. These legacy trees represent a critical carbon reservoir within cocoa fields and are essential for maintaining structural diversity and supporting regeneration processes. Spontaneous trees, while contributing less to total field-level carbon stocks, add meaningful amounts and play an important complementary role in sustaining agroforestry resilience.

(2) At the individual tree level, we observed that spontaneous trees accumulate more carbon from growth over time than planted trees, indicating their higher carbon sequestration efficiency. This suggests that encouraging the growth and survival of spontaneously established trees could be more effective for increasing biomass accumulation and carbon sequestration in the medium term than relying solely on planted individuals.

(3) Regarding socio-environmental drivers, two key factors emerged: farmers' training and land tenure security. Farmers with greater botanical knowledge were more likely to preserve remnant trees and allow spontaneous trees to grow. Likewise, land ownership significantly influenced carbon outcomes, as landowners were more inclined to invest in long-term tree management practices, including the retention and nurturing of naturally regenerating trees.

Together, these findings emphasize that optimizing carbon storage in cocoa agroforestry systems requires more than just planting efforts. Instead, it demands an integrated strategy that values and supports remnant and spontaneous trees,

enhances farmers' awareness and technical capacity, and ensures secure land tenure. These elements are critical to enabling rural communities in West Africa to contribute meaningfully to climate change mitigation while sustaining productive agroforestry landscapes. Additionally, future research should assess the carbon sequestration capacity of both planted and spontaneous trees at the population level, taking into account tree mortality and population dynamics. Such studies would contribute to a more comprehensive understanding of the carbon balance in cocoa agroforestry systems and provide deeper insights into their long-term role in climate change mitigation.

## Supporting information

**S1 Appendix. Full list of botanical names per cohort.**
(DOCX)

**S2 Appendix. Distribution of DBH between tree cohorts.**
(TIFF)

**S3 Appendix. Distribution of Height between tree cohorts.**
(TIFF)

**S4 Appendix. Density distribution of Wood Density between cohorts.**
(TIFF)

## Author contributions

Conceptualization: Isaac Kouamé Konan, Anny Estelle N'Guessan, Justin Kassi N'Dja, Bruno Hérault.

Data curation: Isaac Kouamé Konan, Aimé Kouassi, Marie Ruth Dago, Raphaël Aussenac, Stéphane Guei.

Formal analysis: Isaac Kouamé Konan, Bruno Hérault.

Funding acquisition: Patrick Jagoret.

Investigation: Isaac Kouamé Konan, Anny Estelle N'Guessan, Bruno Hérault.

Methodology: Isaac Kouamé Konan, Anny Estelle N'Guessan, Bruno Hérault.

Project administration: Anny Estelle N'Guessan, Bruno Hérault.

Resources: Bruno Hérault.

Software: Isaac Kouamé Konan, Bruno Hérault.

Supervision: Anny Estelle N'Guessan, Justin Kassi N'Dja, Bruno Hérault.

Validation: Anny Estelle N'Guessan, Justin Kassi N'Dja, Bruno Hérault.

Visualization: Isaac Kouamé Konan.

Writing – original draft: Isaac Kouamé Konan, Bruno Hérault.

Writing – review & editing: Anny Estelle N'Guessan, Aimé Kouassi, Marie Ruth Dago, Justin Kassi N'Dja, Raphaël Aussenac, Stéphane Guei, Patrick Jagoret.

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
