## [Decision Letter · Decision Letter 0]

PONE-D-25-02686Unsung Climate Guardians: The Overlooked Role of Remnant and Spontaneous Trees in Carbon Stocks and Fluxes in West African Cocoa FieldsPLOS ONE

Dear Dr. Bruno Hérault

Thank you for submitting your manuscript to PLOS ONE. After careful consideration, we feel that it has merit but does not fully meet PLOS ONE’s publication criteria as it currently stands. Therefore, we invite you to submit a revised version of the manuscript that addresses the points raised during the review process.

Dear authors, thank you for submitting your manuscript "Unsung Climate Guardians: The Overlooked Role of Remnant and Spontaneous Trees in Carbon Stocks and Fluxes in West African Cocoa Fields." Your research represents a valuable contribution to understanding carbon dynamics in cocoa agroforestry systems. However, several revisions are needed to enhance the manuscript's scientific rigor. Regarding technical aspects, it is necessary to specify the conversion factor used for aboveground biomass to carbon content, detail the allometric equations for each species, and address the unavailability of the BIOMASS package on CRAN by suggesting alternatives or justifying the methodology used. Concerning the study design, it is important to clearly establish the selection criteria for the 15 studied locations, explain the procedure for demarcating the 1000 m² area, and elaborate on the methodology for assessing farmers' botanical knowledge, specifying whether visual aids or local names were used for species identification. The statistical analysis could be strengthened by including tree density disaggregation by categories (planted, remnant, and spontaneous), adding species richness analysis by cohort, and providing detailed justification for using Bayesian analysis. The presentation of results requires including carbon stock and flux ranges by tree origin in the abstract, developing a comprehensive table with descriptive statistics by cohort, and adding a conceptual diagram illustrating the factors affecting carbon stocks and fluxes. In the discussion, it is important to delve deeper into the analysis of wood density's role in carbon sequestration, further discuss factors favoring natural regeneration, especially regarding land tenure, and acknowledge methodological limitations in determining spontaneous trees' age. Regarding minor technical corrections, the format of species names should be revised (remove italics from author names), include author names and APG system in binomial nomenclature, and clarify production units. The manuscript shows promise and addresses an important topic in agroforestry and climate science. Implementation of these revisions will significantly strengthen your work and prepare it for publication.

We look forward to receiving your revised manuscript.

Kind regards,

Juan Carlos Suárez Salazar

Academic Editor

PLOS ONE

Journal Requirements:

2. Please include a complete copy of PLOS’ questionnaire on inclusivity in global research in your revised manuscript. Our policy for research in this area aims to improve transparency in the reporting of research performed outside of researchers’ own country or community. The policy applies to researchers who have travelled to a different country to conduct research, research with Indigenous populations or their lands, and research on cultural artefacts. The questionnaire can also be requested at the journal’s discretion for any other submissions, even if these conditions are not met.  Please find more information on the policy and a link to download a blank copy of the questionnaire here: https://journals.plos.org/plosone/s/best-practices-in-research-reporting. Please upload a completed version of your questionnaire as Supporting Information when you resubmit your manuscript.”

4. Please note that your Data Availability Statement is currently missing [the repository name and/or the DOI/accession number of each dataset OR a direct link to access each database]. If your manuscript is accepted for publication, you will be asked to provide these details on a very short timeline. We therefore suggest that you provide this information now, though we will not hold up the peer review process if you are unable.

“This study was conducted within the framework of the Cocoa4Future (C4F) project, which is funded by the European DeSIRA Initiative under grant agreement No. FOOD/2019/412-132 and by the French Development Agency.”

“This study was conducted within the framework of the Cocoa4Future (C4F) project, which is funded by the European DeSIRA Initiative under grant agreement No. FOOD/2019/412-132 and by the French Development Agency. All authors benefited from this funding.”

6. We note that Figure 1 in your submission contain [map/satellite] images which may be copyrighted. All PLOS content is published under the Creative Commons Attribution License (CC BY 4.0), which means that the manuscript, images, and Supporting Information files will be freely available online, and any third party is permitted to access, download, copy, distribute, and use these materials in any way, even commercially, with proper attribution. For these reasons, we cannot publish previously copyrighted maps or satellite images created using proprietary data, such as Google software (Google Maps, Street View, and Earth). For more information, see our copyright guidelines: http://journals.plos.org/plosone/s/licenses-and-copyright.

Additional Editor Comments (if provided):

The manuscript "Unsung Climate Guardians: The Overlooked Role of Remnant and Spontaneous Trees in Carbon Stocks and Fluxes in West African Cocoa Fields" presents original and relevant research on carbon dynamics in cocoa fields in Côte d'Ivoire. While the work demonstrates significant strengths in its general structure, well-developed introduction, and presentation of results, there are methodological and analytical aspects that require attention before publication.

The primary concern lies in technical aspects related to biomass and carbon estimation. It is necessary to specify the conversion factor used to transform aboveground biomass into carbon content, as well as detail the allometric equations employed for each species. Additionally, the issue of the BIOMASS package, currently unavailable on CRAN, needs to be addressed, suggesting alternatives or justifying the methodology used.

Regarding study design, it is essential to explicitly state the selection criteria for the 15 studied locations and detail the procedure for demarcating the 1000 m² area. The methodology for assessing farmers' botanical knowledge also requires further elaboration, specifying whether visual aids or local names were used in the species identification process.

The statistical analysis could be strengthened by disaggregating tree density by categories (planted, remnant, and spontaneous) and including species richness analysis by cohort. The justification for using Bayesian analysis and its advantages over conventional methods needs to be elaborated in greater detail.

The presentation of results needs to be enriched with additional information. It is suggested to include in the abstract the ranges of carbon stock and fluxes according to tree origin, as well as develop a comprehensive table with descriptive statistics by cohort. A conceptual diagram illustrating the factors affecting carbon stocks and fluxes would significantly improve the understanding of the work.

In the discussion, it is important to delve deeper into the analysis of wood density's role in carbon sequestration, particularly concerning the differences observed between planted and spontaneous trees. Factors favoring natural regeneration deserve more attention, especially in the context of land tenure. Furthermore, the methodological limitation in determining spontaneous trees' age and its possible impact on carbon flux estimates should be explicitly acknowledged.

Minor but important technical aspects include correcting the format of species names (author names should not appear in italics), including author names and APG system in binomial nomenclature, and clarifying production units.

In conclusion, although the manuscript requires moderate revisions, it presents a valuable contribution to understanding carbon dynamics in cocoa agroforestry systems. The suggested modifications, especially in methodological aspects and results presentation, will significantly strengthen the scientific rigor and clarity of the work. Once these improvements are implemented, the manuscript will be in appropriate condition for publication, making a significant contribution to knowledge about the role of spontaneous and remnant trees in carbon storage and flux in tropical agricultural systems.

Reviewers' comments:

Reviewer's Responses to Questions

**Comments to the Author**

1. Is the manuscript technically sound, and do the data support the conclusions?

Reviewer #1: Yes

Reviewer #2: Yes

Reviewer #3: Yes

2. Has the statistical analysis been performed appropriately and rigorously? 

Reviewer #1: Yes

Reviewer #2: Yes

Reviewer #3: Yes

3. Have the authors made all data underlying the findings in their manuscript fully available?

Reviewer #1: Yes

Reviewer #2: No

Reviewer #3: No

4. Is the manuscript presented in an intelligible fashion and written in standard English?

Reviewer #1: Yes

Reviewer #2: Yes

Reviewer #3: Yes

5. Review Comments to the Author

Reviewer #1: The BIOMASS package used for estimating carbon content of the trees appears to have been removed from CRAN. Additionally, the authors should specify the conversion factor applied for converting aboveground biomass (AGB) into carbon content.

The Materials and Methods section does not explain how AGB for the trees was estimated. Although the discussion section mentions the use of allometric equations, it would be prudent to explicitly disclose the source of the allometric equations used for different species in the methodology section.

The finding that planted trees typically have lower carbon sequestration could be linked to the fact that many of these species are horticultural, which tend to have lower wood density. It is recommended that the authors investigate the role of wood density further to identify which cohort exhibits the highest density.

The lower number of planted trees across various species, compared to the spontaneous cohort, suggests abundant regeneration potential. The authors could explore factors favoring regeneration, especially in the context of land ownership.

The methodology section regarding carbon flux and the use of Bayesian analysis needs further elaboration. The authors should explain its purpose and highlight the advantages of this methodology compared to conventional approaches.

Reviewer #2: The article is well-written and covers each section in depth. The abstract, introduction, and methodology are consistent with the objectives and research questions developed. The results and discussion are presented in an organized and comprehensive manner. The conclusions summarize the answers to the questions posed in a concise manner.

Reviewer #3: The manuscript “PONE-D-25-02686” titled “Unsung Climate Guardians: The Overlooked Role of Remnant and Spontaneous Trees in Carbon Stocks and Fluxes in West African Cocoa Fields” present original research which investigated carbon dynamics across cocoa fields, in Côte d'Ivoire. Authors (i) estimated carbon stock levels of planted, spontaneous, and remnant trees in cocoa farms, (ii) assessed the relative contribution of planted, and spontaneous trees to carbon fluxes within cocoa farms, and (iii) identified the key socio-environmental factors influencing carbon stocks and fluxes in these systems. This is an interesting contribution in understanding the carbon stock and fluxes in cocoa fields and their determinants considering that previous studies have overlooked spontaneous, and remnant trees in estimating carbon stock and fluxes in cocoa fields.

The manuscript is overall well written and clear. However, there are some aspects that need to be improved before the manuscript can be accepted for publication.

The introduction is well written, and the methodology is overall correct and reproductible. The data analyses are also acceptable but can be improved. The results are well presented and discussed, but additional results are needed (from the data of the authors) to support some of the discussion.

Below the specific concerns.

Abstract ------

It would be informative to add the range of the carbon stock and carbon according to tree origin.

It would also be informative to add the relative contribution of the significant factors affecting carbon stock and flux.

Introduction ------

Well written

• Line 101 : what is the added value of using both identity and assess and not only one, and how this translate in the results?

• A conceptual diagram illustrating the effect (negative, positive, or neutral or mixed) of these factors (especially the eight in table 1) would increase the paper clarity. This could be added either at the end of the introduction or in the methodology section.

Study sites -----

• Line 110 : Authors selected 15 locations without explaining how the selection was made. How were these sites selected? Were there any criteria used to select these locations ? Authors should expand this point.

• Line 118. Is it per ha ? or it is the total production.

• Line 119 : How do you make sure that this declaration is accurate, as age was an important component of the carbon flux model ?

• Line 131 : the authors estimated botanical knowledge of the farmers based on the 23 most common species in the area. They should expand more on how this was measured. For example, did they show the pictures of the species to the farmers, or did they give him/her the local names of the species?

• Line 135 : Cocoa density was found as an important variable influencing both carbon stock and flux. It would be interesting in addition to this, to disaggregate tree density in tree density of planted trees, tree density of remnant trees, and tree density of spontaneous trees to disentangle the role of each. Furthermore, the total species richness, disaggregated in species richness of spontaneous trees, and species richness of remnant trees would provide valuable insights in the analyses, and the authors have the data to do so.

• Line 136 : Authors are invited to expand more on how the 1000 m2 area was demarcated in each cocoa field.

Results ------

• Lines 181 – 187 : It would be good to put this under the subsection « Tree density and composition ». In this sub-section it would also be valuable to describe tree diversity overall and per cohort, and on average per plot, at least as a supplementary data.

• Lines 189 – 196 : In addition to figure 2, it would be good to add a table summarising the carbon stock in terms of min ; mean, max, Q1, Q2, and Q3, and cv (%) per cohort. The Figure 2 only is not sufficient to get a comprehensive picture of the data presented.

• Lines 247 – 257 : Species authors’ names should not be italicised.

Discussion -----

• Lines 253 : While I agree with that discussion based on some previous evidence, I believe that it would be good and useful that authors report on species diversity in each cohort based on their field data.

• Table 2 : It would be useful to add the authors of each species and the APG followed in the binomial naming of the species. Authors could also add the full list of species recorded in each tree cohort as a supplementary data.

• Lines 274 – 276 : As you have the data at hand, it would be preferable to plot the relationship between level of botanical knowledge and tree density and diversity. This is far better as evidence to support the findings and discussion.

• Lines 300 – 301 : I agree with this. But one point to raise is that the farmers may underestimate the age of the spontaneous trees, and this could have overestimated the carbon flux for these trees. This is a major point to acknowledge as the age was not determined with a rigorous scientific approach.

6. PLOS authors have the option to publish the peer review history of their article (what does this mean? ). If published, this will include your full peer review and any attached files.

**Do you want your identity to be public for this peer review?** For information about this choice, including consent withdrawal, please see our Privacy Policy .

Reviewer #1: **Yes: ** Suresh Ramanan S

Reviewer #2: No

Reviewer #3: **Yes: ** Kolawole Valere Salako

---

## [Decision Letter · Decision Letter 1]

PONE-D-25-02686R1Unsung Climate Guardians: The Overlooked Role of Remnant and Spontaneous Trees in Carbon Stocks and Fluxes in West African Cocoa FieldsPLOS ONE

Dear Dr. Hérault,

Thank you for submitting your manuscript to PLOS ONE. After careful consideration, we feel that it has merit but does not fully meet PLOS ONE’s publication criteria as it currently stands. Therefore, we invite you to submit a revised version of the manuscript that addresses the points raised during the review process.

We look forward to receiving your revised manuscript.

Kind regards,

Juan Carlos Suárez Salazar

Academic Editor

PLOS ONE

Journal Requirements:

Additional Editor Comments:

The manuscript titled "Unsung Climate Guardians: The Overlooked Role of Remnant and Spontaneous Trees in Carbon Stocks and Fluxes in West African Cocoa Fields" provides a detailed review of biomass distribution among remnant, regenerating, and planted trees in cocoa agroforestry systems, based on farmer surveys. This study highlights the importance of remnant trees and suggests faster carbon accumulation in regenerating trees compared to planted ones, while also offering interesting insights into farmer knowledge and previous land use.

Reviewer comments emphasize the contribution of the study, though they suggest improvements in several areas for better clarity. It is recommended to improve wording in certain lines and clarify methodological procedures, such as the calculation of tree height and the acquisition of wood density. The use of refined carbon fractions is suggested, and the growth gain analysis should be adjusted to younger trees due to the scarcity of planted trees over 30 years old.

In response to previous reviews, the manuscript's update to comply with PLOS ONE's style requirements and the inclusion of an inclusivity questionnaire are appreciated. Methodological justifications and the use of Bayesian analysis have been clarified, strengthening the study's robustness. The expanded discussion on wood density and factors of natural regeneration, especially in the context of land tenure, adds significant value to the manuscript. Including a DOI for the data availability statement ensures transparency and accessibility, which are crucial for the study's reproducibility.

Reviewers' comments:

Reviewer's Responses to Questions

**Comments to the Author**

1. If the authors have adequately addressed your comments raised in a previous round of review and you feel that this manuscript is now acceptable for publication, you may indicate that here to bypass the “Comments to the Author” section, enter your conflict of interest statement in the “Confidential to Editor” section, and submit your "Accept" recommendation.

Reviewer #4: (No Response)

Reviewer #5: (No Response)

2. Is the manuscript technically sound, and do the data support the conclusions?

Reviewer #4: No

Reviewer #5: Yes

3. Has the statistical analysis been performed appropriately and rigorously? 

Reviewer #4: No

Reviewer #5: Yes

4. Have the authors made all data underlying the findings in their manuscript fully available?

Reviewer #4: Yes

Reviewer #5: Yes

5. Is the manuscript presented in an intelligible fashion and written in standard English?

Reviewer #4: Yes

Reviewer #5: Yes

6. Review Comments to the Author

Reviewer #4: The manuscript tittle "Unsung Climate Guardians: The Overlooked Role of Remnant and Spontaneous Trees

in Carbon Stocks and Fluxes in West African Cocoa Fields" is of general interest for agroforestry researchers and practitioners, yet in the current form, it has five main limitations:

1) Within the title the work Fluxes is not of correct use here, the authors are simply projecting carbons sequestration potential of a classes of trees on farms. Fluxes mean that you are including population dynamic in the analysis.

2) The historical climate records you are using for the analysis has a GAP which might heavily affect you analysis and projection, precipitation and temperature records are available only from 1979 to 2013, so how you project the remaining climate records to account for sequestration potential of subjects trees?.

3) The analysis and assumption the authors used in the study is no entirely correct, the authors are analysis and projecting on farm tree population as "Stable" population, no changes are foreseen, but this is incorrect, trees on farms (specially the spontaneous) experience recruitment rates, mortality rate and harvest rates, none of this aspect are considered in the analysis not even in the discussion section, therefore the projection of carbon sequestration potential are not valid. The authors need to include the tree management aspect into the analysis/discussion.

4) Results from Figure 3 are quite confusing, growth patter of trees usually follow a S shape curve which means tree growth rates decreases overtime until a plateau is reached, therefore trees sequestration potential decreases overtime, however, figure 3 show low or even decreasing sequestration rate, quite the opposite of spontaneous trees, this findings require eighter a review of the growth model used (the allometric equations that is useful for the dbh range of inventory trees) for this particular tree class or a well references discussion quite far for "nursery issues"

5) the discussion is out of focus, there is a lack of discussion regarding population dynamic of the guardian trees in cacao farms as well as the potential effects of elevated C02 on overall tree growth. The conclusion is quite speculative, in covers topics such as seed dispersal, natural regeneration, farmers training, etc and does not answers the three initial research questions. Conclusion is overall redundant, speculative and vague.

Reviewer #5: This manuscript surveys trees in cocoa agroforestry and partitions biomass between remnant, regenerating and planted trees informed by surveys with farmers. This reinforces the importance of remnant trees, but also supports faster carbon accumulation from regenerating rather than planted trees. There are also interesting insights into the effects of farmer knowledge and prior land-use.

This is an interesting contribution, although I have a number of points that could do with more clarity. I could not see a response to reviewers, so apologise if I have suggested something that was rebutted previously.

Detailed comments:

Line 37-38 – Awkward phrasing – better as “Carbon stocks were higher where farmers had greater knowledge and where there was previous forest land use”?

Throughout – I’d personally talk about passively regenerating trees (or naturally regenerating trees) rather than spontaneous trees, as I find the latter wording a bit strange, but it is clearly defined so up to the authors.

82 – slow growth rather than stunt growth, as I think this is saying low precipitation slows growth rates rather than diminishes tree final size.

Line 128 – How many ha were surveyed?

Line 178 – Probably not worth redoing all the analysis for, but would be better to use one of the refined carbon fractions from Thomas and Martin 2018 rather than the IPCC default.

Line 178 – Which procedure was used to estimate tree height? The environmental stress parameter? Also define how wood density was obtained (taxonomic lookup through getWD I presume, if so state matches at species and genus level, and if family or plot-level values were used for taxa not matched at genus level).

Line 181 – This isn’t net carbon flux, it is carbon flux due to growth, and potentially would be better named as carbon gains or similar. Within a plot, there will be carbon fluxes in due to growth and out due to mortality, including unobserved growth of trees that have now died. What is looked at here is one part of this – fluxes in due to growth of the cohort of surviving trees. This needs to be more tightly defined.

Line 195 – It would be helpful to explain (in words rather than an equation) what coefficients are free to vary with what in this model. If I understand right, there is a power law effect of age which varies between cohorts, and observed biomass separately varies with a set of socio-ecological variables. So age-specific effects can vary between plots, but do not explicitly interact with socio-ecological factors? Also, why are age power-law effects allowed to vary if they are constrained by metabolic and geometric scaling? Is it because these are size-scaling factors, and you are capturing the varying age-size relationship?

Line 207-208 – While there are advantages with this modelling framework, the biggest uncertainty is the likely large errors in farmer estimates of tree age. This is not accounted for in this modelling framework nor a simple regression, and it needs to be clearly stated that there is a big uncertainty which is not accounted for. I know this is talked about in the discussion, but mention here as well.

Line215-224 – I’d find it helpful to have some boxplots (e.g. size class, wood density) between the groups.

Line 240 – Watch use of European decimal (, vs .).

Line 253 – Define what the plus minus values are showing.

Line 294-301 – I think the discussion of remnant trees can be improved. It should distinguish between tree species identify (e.g. are dense wooded species left as they are hard to cut [no according to non-significant differences in wood density]) and the more obvious they store more carbon because they are old big trees rather than young ones.

Figure 3. There are very few planted trees more than ~30 years old, so maybe restrict the analysis of growth gains to the younger trees.

Figure 4/5. Caution is needed when inferring causality about farmer knowledge. Could farmers with more diverse agroforests (or living closer to forests) know more tree species?

7. PLOS authors have the option to publish the peer review history of their article (what does this mean? ). If published, this will include your full peer review and any attached files.

**Do you want your identity to be public for this peer review?** For information about this choice, including consent withdrawal, please see our Privacy Policy .

Reviewer #4: **Yes: ** Luis Orozco Aguilar

Reviewer #5: No

---

## [Author Response · Author response to Decision Letter 2]

10 Jun 2025

Response to reviewers downloaded with the cover letter.

---

## [Editor Report · Decision Letter 2]

Unsung climate guardians: the overlooked role of remnant and spontaneous trees in carbon stocks and gains from tree growth in West African cocoa fields

PONE-D-25-02686R2

Dear Dr. Bruno Hérault,

We’re pleased to inform you that your manuscript has been judged scientifically suitable for publication and will be formally accepted for publication once it meets all outstanding technical requirements.

Kind regards,

Juan Carlos Suárez Salazar

Academic Editor

PLOS ONE

Additional Editor Comments (optional):

Analysis of the document shows that the authors made substantial modifications that significantly strengthened the manuscript in response to the evaluations received. The changes included important adjustments in terminology, replacing the term "fluxes" with "growth gains" throughout the document, including the title. The use of historical climate data (1979-2013) was also clarified, explaining that the study does not intend to project future climate impacts but rather analyze spatial variability based on existing historical records.

The concern about the assumption of a stable tree population was addressed by clarifying that the analysis focuses on current stocks and individual gains, without attempting to model complete population dynamics. Regarding the growth patterns presented in Figure 3, a more detailed explanation based on metabolic scaling theory was provided, and residual plots were included to validate the model.

Improvements in text clarity included better definition of technical terms and specification of the total area studied (240.5 ha). Methodological adjustments incorporated clarification of the tree height estimation procedure and wood density acquisition. Additionally, the analysis was limited to trees under 40 years old for greater precision.

Data visualization was enhanced with the addition of box plots for size classes and wood density, including new appendices (2-4). The authors also explicitly acknowledged the study's limitations, particularly regarding uncertainty in age estimates and interpretation of farmers' botanical knowledge.

The conclusion was restructured to directly address the three main research questions, removing speculative elements and emphasizing empirical findings. Overall, the modifications maintain the central contribution of the study on the role of different types of trees in cocoa agroforestry systems while effectively addressing all major concerns raised in the initial evaluation.
---

## [Editor Report · Acceptance letter]

PONE-D-25-02686R2

PLOS ONE

Dear Dr. Hérault,

I'm pleased to inform you that your manuscript has been deemed suitable for publication in PLOS ONE. Congratulations! Your manuscript is now being handed over to our production team.

Kind regards,

on behalf of

Dr. Juan Carlos Suárez Salazar

Academic Editor

PLOS ONE